# Positively Weighted Kernel Quadrature via Subsampling

**Satoshi Hayakawa,  Harald Oberhauser,  Terry Lyons**
Mathematical Institute, University of Oxford
{hayakawa,oberhauser,tlyons}@maths.ox.ac.uk

## Abstract

We study kernel quadrature rules with convex weights. Our approach combines the spectral properties of the kernel with recombination results about point measures. This results in effective algorithms that construct convex quadrature rules using only access to i.i.d. samples from the underlying measure and evaluation of the kernel and that result in a small worst-case error. In addition to our theoretical results and the benefits resulting from convex weights, our experiments indicate that this construction can compete with the optimal bounds in well-known examples. [1]

## 1 Introduction

The goal of numerical quadrature is to provide, for a given probability measure $\mu$ on a space $\mathcal{X}$, a set of points $x_1, \ldots, x_n \in \mathcal{X}$ and weights $w_1, \ldots, w_n \in \mathbb{R}$ such that

$$\sum_{i=1}^{n} w_i f(x_i) \approx \int_{\mathcal{X}} f(x) \, \mathrm{d}\mu(x) \tag{1}$$

holds for a large class of functions $f : \mathcal{X} \to \mathbb{R}$. Kernel quadrature focuses on the case when the function class forms a reproducing kernel Hilbert space (RKHS). What makes kernel quadrature attractive, is that the kernel choice provides a simple and flexible way to encode the regularity properties of a function class. Exploiting such regularity properties is essential when the integration domain $\mathcal{X}$ is high-dimensional or the function class is large. Additionally, the domain $\mathcal{X}$ does not have to be Euclidean, but can be any topological space that carries a positive semi-definite kernel.

More formally, given a set $Q := \{(w_i, x_i) : i = 1, \ldots, n\} \subset \mathbb{R} \times \mathcal{X}$ denote with $\mu^Q := \sum_{i=1}^{n} w_i \delta_{x_i}$ the resulting measure on $\mathcal{X}$. We refer to $Q$ [resp. $\mu^Q$] as a quadrature [resp. quadrature measure], to the points $x_1, \ldots, x_n$ as the support of $Q$ [resp. $\mu^Q$]. The aim of kernel quadrature is to construct quadrature measures $\mu^Q$ that have a small worst-case error

$$\mathrm{wce}(Q; \mathcal{H}_k, \mu) := \sup_{\|f\|_{\mathcal{H}_k} \leq 1} \left| \int_{\mathcal{X}} f(x) \, \mathrm{d}\mu^Q(x) - \int_{\mathcal{X}} f(x) \, \mathrm{d}\mu(x) \right|, \tag{2}$$

where $\mathcal{H}_k$ denotes the RKHS associated with a positive semi-definite kernel $k$. If the weights are positive and sum up to one, $w_i > 0$, $\sum_{i=1}^{n} w_i = 1$, then we refer to $Q$ as a convex quadrature rule.

**Contribution.**    The primary contribution of this article is to leverage recombination (a consequence of Carathéodory's Theorem) with spectral analysis of kernels to construct convex kernel quadrature rules and derive convergence rates. We also provide efficient algorithms that compute these quadrature rules; they only need access to i.i.d. samples from $\mu$ and the evaluation of the kernel $k$. See Table 1 for a comparision with other kernel quadrature constructions.

---

[1]Code: https://github.com/satoshi-hayakawa/kernel-quadrature

36th Conference on Neural Information Processing Systems (NeurIPS 2022).

The table is written by using $\sigma_n$ and $r_n$, which represent a sort of decay of the kernel with respect to $\mu$. Typical regimes are $\sigma_n \sim n^{-\beta}$ (e.g. Sobolev) or $\sigma_n \sim \exp(-\gamma n)$ (e.g. Gaussian) depending on the 'smoothness' of the kernel [e.g. 16, 3] (see also Section B.3), and in such regimes (with $\beta \geq 2$ or $\gamma > 0$), $\sigma_n$ or $r_n (\lesssim n\sigma_n)$ provide faster rates than $\mathrm{wce}^2 \sim 1/n$ of the usual Monte Carlo rate. For more examples including multivariate Sobolev spaces, see Bach [3, Section 2.3].

**Limitation.** Our proposed methods are based on either Mercer or Nyström approximation. Though our Mercer-based methods result in strong theoretical bounds, they require the knowledge of Mercer decomposition like [3, 7, 8], which is not available for general $(k, \mu)$. Our Nyström-based methods apply to much more general situations and outperform existing methods in experiments, but the $n/\sqrt{\ell}$ term makes their theoretical bound far from competitive. Further study is needed to bridge the gap between theory and empirical results.

| Method | Bound of squared wce | Computational complexity | C | M | E |
|---|---|---|---|---|---|
| Herding [10, 4] | $1/n$ | $n \cdot (n$: global optimization$)$ | ✓ | ✓ | |
| SBQ [25] | Not found | $n \cdot (n^2$: global optimization$)$ | | ✓ | |
| Leveraged [3] | $\sigma_m, m = \mathcal{O}(n \log n)$ | Unavailable | | | |
| DPP [7, 6] | $r_{n+1}$ | $n^3 \cdot ($rejection sampling$)$ | | | |
| CVS [8] | $\sigma_{n+1}$ | Unavailable | | ✓ | |
| KT++ [14, 15, 56] | $(1/n^2 + 1/N)\operatorname{polylog}(N)$ | $N \log^3 N$ | ✓ | ✓ | ✓ |
| Ours: | | | | | |
| Mercer$^\dagger$ | $r_n$ | $nN_\varphi + C(n, N_\varphi)$ | ✓ | | |
| M. + empirical$^\ddagger$ | $r_n + \frac{1}{N}$ | $nN + n^3 \log(N/n)$ | ✓ | | ✓ |
| Nyström$^\dagger$ | $n\sigma_n + r_{n+1} + \frac{n}{\sqrt{\ell}}$ | $n\ell N_\varphi + n\ell^2 + C(n, N_\varphi)$ | ✓ | ✓ | |
| N. + empirical$^\ddagger$ | $n\sigma_n + r_{n+1} + \frac{n}{\sqrt{\ell}} + \frac{1}{N}$ | $n\ell N + n\ell^2 + n^3 \log(N/n)$ | ✓ | ✓ | ✓ |

Table 1: Comparison on $n$-point kernel quadrature rules. We are omitting the $\mathcal{O}$ notation throughout the table. Note that the assumption under which the theoretical guarantee holds varies from method to method, and this table displays just a representative bound derived in the cited references. Here are remarks on the notation. (1) $\sigma_m$ is the $m$-th eigenvalue of the integral operator $\mathcal{K}$, and $r_m = \sum_{i=m}^{\infty} \sigma_i$. (2) The symbols in the first line respectively mean C: convex, M: *not* using the knowledge of Mercer decomposition, and E: *not* using the knowledge of expectations such as $\int_{\mathcal{X}} k(x, y) \, d\mu(y)$. (3) The ($m$: global optimization) is indicating the cost of globally optimizing a function whose evaluation costs $\Theta(m)$. (†) Mercer/Nyström are the algorithms based on random convex hulls, see Section 2.4 and Appendix D. (‡) M./N. + empirical are the algorithms discussed in the main text.

**Why Convex Weights?** There are several reasons why convex weights are preferable: (i) **Positive Integral Operator:** Kernel quadrature provides an approximation of the integration operator $f \mapsto I(f) = \int f(x) \, d\mu(x)$. Hence, a natural requirement is to preserve basic properties of this operator and positive weights preserve the positivity of this operator. (ii) **Uniform estimates and Robustness:** In applications, the RKHS $\mathcal{H}_k$ may be mis-specified if a quadrature rule with negative weights is applied to a function $f \notin \mathcal{H}_k$, the approximation error (1) can get arbitrary bad; in contrast, a simple estimate shows that convex weights give uniform bounds, see Appendix B.4. (iii) **Iteration:** Consider the $m$-fold product of quadrature formulas for approximating $\mu^{\otimes m}$ on $\mathcal{X}^m$. This is a common construction for a multidimensional quadrature formulas (e.g., for polynomials) from one-dimensional formulas [61] or numerics for stochastic differential equations [42]. In doing so, working with a probability measure is strongly preferred, since otherwise the total variation of their $m$-fold product gets exponentially large as $m$ increases ($\|\mu^{\otimes m}\|_{\mathrm{TV}} = \|\mu\|_{\mathrm{TV}}^m$).

**Related Literature.** Roughly speaking, there have been two approaches to kernel-based quadrature formulas: kernel herding and random sampling. In kernel herding or its variants, the points $(x_i)_{i=1}^n$ are found iteratively, typically based on the Frank–Wolfe gradient descent algorithm [10, 4, 25].

In the random sampling approach, $(x_i)_{i=1}^n$ are sampled and subsequently the weights are optimized. Generically, this results only in a signed measure $\mu^Q$ but not a probability measure. Bach [3] and

Belhadji et al. [7] use the eigenvalues and the eigenfunctions of the integral operator $\mathcal{K} : f \mapsto \int_{\mathcal{X}} k(\cdot, y) f(y) \, d\mu(y)$ to obtain a Mercer-type decomposition of $k$ [59]. Bach [3] then uses the eigenvalues and eigenfunctions of $\mathcal{K}$ to define an optimized measure from which the points $(x_i)$ are i.i.d. sampled. This achieves a near optimal rate, but the exact sampling from this measure is usually unavailable, although for special cases, it can be done efficiently. In contrast, Belhadji et al. [7] proposes non-independent sampling based on the determinantal point process [DPP; 24]. These two papers also treat the more general quadrature problem that includes a weight function $g \in L^2(\mu)$, i.e., approximating $\int_{\mathcal{X}} f(x) g(x) \, d\mu(x)$ for $f \in \mathcal{H}_k \subset L^2(\mu)$, which we do not discuss in this paper. Another recently introduced method is kernel thinning [14, 15], which aims at efficient compression of empirical measures that can be obtained by sampling like our '+ empirical' methods. Its acceleration [56] makes it a competitive candidate in terms of compressing $N \sim n^2$ points ('KT++' in Table 1).

Finally, we emphasize that the kernel quadrature literature is vast, and the distinction between herding and sampling is only a rough dichotomy, see e.g. [12, 40, 9, 27, 48, 29, 28, 57]. Beyond kernel quadrature, our algorithms can also contribute to the density estimation approach in [64] which relies on recombination based on Fourier features although we do not pursue this further in this article.

**Outline.** Section 2 contains our main theoretical and methodological contribution. Section 3 provides numerical experiments on common benchmarks. The Appendix contains several extensions of our main result, proofs, and further experiments and benchmarks.

## 2 Main Result

Assume we are given a set[2] of $n - 1$ functions $\varphi_1, \ldots, \varphi_{n-1} : \mathcal{X} \to \mathbb{R}$ such that their linear combinations well approximate functions in $\mathcal{H}_k$. Then our kernel quadrature problem reduces to the construction of an $n$-point discrete probability measure $\mu^{Q_n} = \sum_{i=1}^{n} w_i \delta_{x_i}$ such that

$$\int_{\mathcal{X}} \varphi_i(x) \, d\mu^{Q_n}(x) = \int_{\mathcal{X}} \varphi_i(x) \, d\mu(x) \quad \text{for every } i = 1, \ldots, n - 1. \tag{3}$$

A simple way to *approximately* construct this $\mu^{Q_n}$ is to first, sample $N \gg n$ points, $(y_i)_{i=1}^{N}$, from $\mu$ such that their empirical measure, $\widetilde{\mu}_N = \frac{1}{N} \sum_{i=1}^{N} \delta_{y_i}$, is a good approximation to $\mu$ in the sense that $\int \varphi_i \, d\widetilde{\mu}_N \approx \int \varphi_i \, d\mu$ for $i = 1, \ldots, n - 1$, and secondly, apply a so-called recombination algorithm (Remark 1) that takes as input $(y_i)_{i=1}^{N}$ and $n$ functions $\varphi_1, \ldots, \varphi_{n-1}$ and outputs a measure $\mu^{Q_n} = \sum w_i \delta_{x_i}$ by selecting a subset $(x_i)_{i=1}^{n}$ of the points $(y_i)_{i=1}^{N}$ and giving them weights $(w_i)_{i=1}^{n}$ such that $\mu^{Q_n}$ is a probability measure that satisfies the equation (3) with $\mu$ replaced by $\widetilde{\mu}_N$.

The challenging parts of this approach are (i) to construct functions $\varphi_1, \ldots, \varphi_{n-1}$ that approximately span the RKHS $\mathcal{H}_k$ for a small $n$; (ii) to arrive at good quantitative bounds despite the (probabilistic) sampling error resulting from the use of the empirical measure $\widetilde{\mu}_N$, and the function approximation error via $\varphi_1, \ldots, \varphi_{n-1}$. To address (i) we look for functions such that

$$k(x, y) \approx k_0(x, y) := \sum_{i=1}^{n-1} c_i \varphi_i(x) \varphi_i(y) \tag{4}$$

with some $c_i \geq 0$. Two classic ways to do this are the Mercer and Nyström approximations. The remaining, item (ii) is our main contribution. Theorem 1 shows that the worst-case error, (5), is controlled by the sum of two terms: the first term stems from the kernel approximation (4), the second term stems from the sample error.

**Theorem 1.** *Let $\mu$ be a Borel probability measure on $\mathcal{X}$ and $k$ a positive semi-definite kernel on $\mathcal{X}$ such that $\int_{\mathcal{X}} k(x, x) \, d\mu(x) < \infty$. Further, let $n$ be a positive integer and assume $k_0$ is a positive semi-definite kernel on $\mathcal{X}$ such that*

   *1. $k - k_0$ is a positive semi-definite kernel on $\mathcal{X}$, and 2. $\dim \mathcal{H}_{k_0} < n$.*

*There exists a function* KQuad *such that if $D_N$ is a set of $N$ i.i.d. samples from $\mu$, then $Q_n = $* KQuad$(D_N)$ *is a random $n$-point convex quadrature that satisfies*

$$\mathbb{E}_{D_N} \left[ \mathrm{wce}(Q_n; \mathcal{H}_k, \mu)^2 \right] \leq 8 \int_{\mathcal{X}} (k(x, x) - k_0(x, x)) \, d\mu(x) + \frac{2 c_{k,\mu}}{N}. \tag{5}$$

---

[2]The number $n - 1$ stems from Carathéodory's theorem, Remark 1, and leads to an $n$ point quadrature rule.

where $c_{k,\mu} := \int_{\mathcal{X}} k(x,x)\,\mathrm{d}\mu(x) - \iint_{\mathcal{X} \times \mathcal{X}} k(x,y)\,\mathrm{d}\mu(x)\,\mathrm{d}\mu(y)$.

*Moreover, the support of $Q_n$ is a subset of $D_N$ and given functions $\varphi_1, \ldots, \varphi_{n-1} \in L^1(\mu)$ with $\mathcal{H}_{k_0} \subset \mathrm{span}\{\varphi_1, \ldots, \varphi_{n-1}\}$, $Q_n = \mathrm{KQuad}(D_N)$ can be computed with Algorithm 1 in $\mathcal{O}\big(nN + n^3 \log(N/n)\big)$ computational steps.*

The function KQuad is deterministic but since $D_N$ is random, the resulting quadrature rule $Q_n$ is random, hence also the resulting worst case error $\mathrm{wce}(Q; \mathcal{H}_k, \mu)$ and the expectation in (5) denotes the expectation over the $N$ samples in $D_N$. The theoretical part of Theorem 1 follows from more general results that we present and prove in the Appendix: Theorem 7 proves the inequality, essentially by comparing $\mathcal{H}_k$ with $\mathcal{H}_{k_0}$; Theorem 8 proves the existence. The algorithmic part of Theorem 1 is discussed in Section 2.1 below. Theorem 1 covers our two main examples for the construction of $k_0$, resp. the choice of $\varphi_1, \ldots, \varphi_{n-1}$, and for which the error estimate gets quite explicit: the Mercer approximation, see Section 2.2, and the Nyström approximation, see Section 2.3. The former requires some knowledge about the spectrum of the kernel which is, however, known for many popular kernels; the latter works in full generality but yields worse theoretical guarantees for the convergence rate. Finally, we emphasize that $N$ and $n$ in Theorem 1 can be chosen independently and we will see that from a computational point the choice $N \sim n^2$ is preferable in which case (5) is faster rate than Monte Carlo, see also Table 1.

## 2.1 Algorithm

---

**Algorithm 1** Kernel Quadrature with Convex Weights via Recombination KQuad

---

**Input:** A positive semi-definite kernel $k$ on $\mathcal{X}$, a probability measure $\mu$ on $\mathcal{X}$, integers $N \geq n \geq 1$, another kernel $k_0$, functions $\varphi_1, \ldots, \varphi_{n-1}$ on $\mathcal{X}$ with $\mathcal{H}_{k_0} \subset \mathrm{span}\{\varphi_1, \ldots, \varphi_{n-1}\}$ and a set $D_N$ of $N$ i.i.d. samples from $\mu$.

**Output:** A set $Q_n := \big\{(w_i, x_i) \mid i = 1, \ldots, n\big\} \subset \mathbb{R} \times \mathcal{X}$ with $w_i \geq 0$, $\sum_{i=1}^{n} w_i = 1$

1: Apply a Recombination Algorithm (Remark 1) with $\boldsymbol{\psi} = (\varphi_1, \ldots, \varphi_{n-1}, k_{1,\mathrm{diag}})^\top$, to the empirical measure $\frac{1}{N} \sum_{y \in D_N} \delta_y$ to obtain points $\{\widetilde{x}_1, \ldots, \widetilde{x}_{n+1}\} \subset D_N$ and weights $\boldsymbol{v} = (v_1, \ldots, v_{n+1})^\top \geq \boldsymbol{0}$ that satisfy $\boldsymbol{1}^\top \boldsymbol{v} = 1$ and $\boldsymbol{\psi}(\widetilde{\boldsymbol{x}})\boldsymbol{v} = \frac{1}{N} \sum_{i=1}^{N} \boldsymbol{\psi}(\widetilde{x}_i)$, where $\boldsymbol{\psi}(\widetilde{\boldsymbol{x}}) = [\boldsymbol{\psi}(\widetilde{x}_1), \ldots, \boldsymbol{\psi}(\widetilde{x}_{n+1})] \in \mathbb{R}^{n \times (n+1)}$.

2: Apply SVD with the matrix $A = [\varphi_{i-1}(y_j)]_{ij} \in \mathbb{R}^{n \times (n+1)}$ with $\varphi_0 = 1$ to find a nonzero vector $\boldsymbol{u} \in \mathbb{R}^{n+1}$ such that $A\boldsymbol{u} = \boldsymbol{0}$ and $k_{1,\mathrm{diag}}(\widetilde{\boldsymbol{x}})^\top \boldsymbol{u} \geq 0$

3: Compute the smallest $\alpha \geq 0$ such that $\boldsymbol{v} - \alpha \boldsymbol{u} \geq \boldsymbol{0}$ and $v_j - \alpha u_j = 0$ for some $j$

4: Return $(w_i)_{i=1}^{n} \leftarrow (v_k - \alpha u_k)_{k \in I}$ and $(x_i)_{i=1}^{n} \leftarrow (\widetilde{x}_k)_{k \in I}$, where $I = \{1, \ldots, n+1\} \setminus \{j\}$

---

Suppose we are given $k_0$ and $\varphi_1, \ldots, \varphi_{n-1} \in L^1(\mu)$ with $\mathcal{H}_{k_0} \subset \mathrm{span}\{\varphi_1, \ldots, \varphi_{n-1}\}$, and also $N$ independent samples from $\mu$ denoted by $D_N = (y_1, \ldots, y_N)$. Theorem 7 in the Appendix shows that if we construct a convex quadrature $Q_n = (w_i, x_i)_{i=1}^{n}$ satisfying

$$\sum_{i=1}^{n} w_i \boldsymbol{\varphi}(x_i) = \frac{1}{N} \sum_{i=1}^{N} \boldsymbol{\varphi}(y_i), \qquad \sum_{i=1}^{n} w_i k_{1,\mathrm{diag}}(x_i) \leq \frac{1}{N} \sum_{i=1}^{N} k_{1,\mathrm{diag}}(y_i), \qquad (6)$$

where $\boldsymbol{\varphi} = (\varphi_1, \ldots, \varphi_{n-1})^\top$ and $k_{1,\mathrm{diag}}(x) = k(x,x) - k_0(x,x)$, it satisfies the bound (5). For this problem, we can use the so-called *recombination* algorithms:

**Remark 1** (Recombination). *Given $d-1$ functions (called test functions) and a probability measure supported on $N > d$ points, there exists a probability measure supported on a subset of $d$ points that gives the same mean to these $d-1$ functions. This follows from Carathéodory's theorem and is known as recombination. Efficient deterministic [38, 43, 61] as well as randomized [11] algorithms exist to compute the new probability measure supported on $d$ points; e.g. deterministic algorithms perform the recombination, step 1, in $\mathcal{O}\big(c_\varphi N + d^3 \log(N/d)\big)$ time, where $c_\varphi$ is the cost of computing all the test functions at one sample. If each function evaluation is in constant time, $c_\varphi = \mathcal{O}(d)$.*

Let us briefly provide the intuition behind the deterministic recombination algorithms. We can solve the problem of "reducing (weighted) $2d$ points to $d$ points in $\mathbb{R}^d$ while keeping the barycenter" by using linear programming or a variant of it. If we apply this to $2d$ points each given by a barycenter

of approximately $\frac{N}{2d}$ points, we can reduce the original problem of size $N$ to a problem of size $d \cdot \frac{N}{2d} = \frac{N}{2}$. By repeating this procedure $\log_2(\frac{N}{d})$ times we obtain the desired measure.

Although the recombination introduced here only treats the equality constraints in (6) we can satisfy the remaining constraints just with $n$ points by modifying it. This is done in Algorithm 1 which works as follows: First, via recombination, find an $(n+1)$-point convex quadrature $R_{n+1} = (v_i, y_i)_{i=1}^{n+1}$ that exactly integrates functions $\varphi_1, \ldots, \varphi_{n-1}, k_{1,\mathrm{diag}}$ with regard to the empirical measure $\frac{1}{N} \sum_{i=1}^{N} \delta_{y_i}$. Second, to reduce one point, find a direction ($-\boldsymbol{u}$ in the algorithm) in the space of weights on $(\widetilde{x}_i)_{i=1}^{n+1}$ that does not change the integrals of $\varphi_1, \ldots, \varphi_{n-1}$ and the constant function 1, and does not increase the integral of $k_{1,\mathrm{diag}}$. Finally, move the weight from $\boldsymbol{v}$ to the above direction until an entry becomes zero, at $\boldsymbol{v} - \alpha\boldsymbol{u}$. Such an $\alpha \geq 0$ exists, as $\boldsymbol{u}$ must have a positive entry since it is a nonzero vector whose entries sum up to one. Now we have a convex weight vector with at most $n$ nonzero entries, so it outputs the desired quadrature satisfying (6).

## 2.2 Mercer Approximation

In this section and Section 2.3, we assume that $k$ has a pointwise convergent Mercer decomposition $k(x,y) = \sum_{m=1}^{\infty} \sigma_m e_m(x)e_m(y)$ with $\sigma_1 \geq \sigma_2 \geq \cdots \geq 0$ and $(e_m)_{m=1}^{\infty} \subset L^2(\mu)$ being orthonormal [59]. If we let $\mathcal{K}$ be the integral operator $L^2(\mu) \to L^2(\mu)$ given by $f \mapsto \int_{\mathcal{X}} k(\cdot, y)f(y)\,\mathrm{d}\mu(y)$, then $(\sigma_m, e_m)_{m=1}^{\infty}$ are the eigenpairs of this operator.

The first choice of the approximate kernel $k_0$ is just the trucation of Mercer decomposition.

**Corollary 2.** *Theorem 1 applied with $k_0(x,y) = \sum_{m=1}^{n-1} \sigma_m e_m(x)e_m(y)$ yields a random convex quadrature rule $Q_n$ such that*

$$\mathbb{E}_{D_N}\left[\mathrm{wce}(Q_n; \mathcal{H}_k, \mu)^2\right] \leq 8\sum_{m=n}^{\infty} \sigma_m + \frac{2c_{k,\mu}}{N}. \tag{7}$$

*Proof.* It suffices to prove the result under the assumption $\int_{\mathcal{X}} k(x,x)\,\mathrm{d}\mu(x) = \sum_{m=1}^{\infty} \sigma_m < \infty$, as otherwise the right-hand side of (7) is infinity.

For $k_1 := k - k_0$, we have that $k_1(x,y) = \sum_{m=n}^{\infty} \sigma_m e_m(x)e_m(y)$ and it is the inner product of $\Phi(x) := (\sqrt{\sigma_m}e_m(x))_{m=n}^{\infty}$ and $\Phi(y)$ in $\ell^2(\{n, n+1, \ldots\})$ and so positive semi-definite. Thus $k$ and $k_0$ satisfies the assumption of Theorem 1, and $\int_{\mathcal{X}} k_1(x,x)\,\mathrm{d}\mu(x) = \sum_{m=n}^{\infty} \sigma_m$ applied to (5) yields the desired inequality. $\square$

## 2.3 Nyström Approximation

Although the Nyström method [68, 13, 34] is primarily used for approximating a large Gram matrix by a low rank matrix, it can also be used for directly approximating the kernel function itself and this is how we use it. Given a set of $\ell$ points $Z = (z_i)_{i=1}^{\ell} \subset \mathcal{X}$, the vanilla Nyström approximation of $k(x,y)$ is given by

$$k(x,y) = \langle k(\cdot,x), k(\cdot,y)\rangle_{\mathcal{H}_k} \approx \langle P_Z k(\cdot,x), P_Z k(\cdot,y)\rangle_{\mathcal{H}_k} =: k^Z(x,y), \tag{8}$$

where $P_Z : \mathcal{H}_k \to \mathcal{H}_k$ is a projection operator onto $\mathrm{span}\{k(\cdot, z_i)\}_{i=1}^{\ell}$. In matrix notation, we have

$$k^Z(x,y) = k(x,Z)W^+ k(Z,y) := [k(x,z_1), \ldots, k(x,z_\ell)]W^+ \begin{bmatrix} k(z_1, y) \\ \vdots \\ k(z_\ell, y) \end{bmatrix}, \tag{9}$$

where $W = (k(z_i, z_j))_{i,j=1}^{\ell}$ is the Gram matrix for $Z$ and $W^+$ denotes its Moore–Penrose inverse. We discuss the equivalence between (8) and (9) in Appendix B.5. As $k^Z$ is an $\ell$-dimensional kernel, there exists an $(\ell+1)$-point quadrature formula that exactly integrates functions in $\mathcal{H}_{k^Z}$. For a quadrature formula, exactly integrating all the functions in $\mathcal{H}_{k^Z}$ is indeed equivalent to exactly integrating $k(z_i, \cdot)$ for all $1 \leq i \leq \ell$, as long as the Gram matrix $k(Z, Z)$ is nonsingular. Proposition 1 in the Appendix provides bound for the associated worst case error. From this viewpoint, the Nyström approximation offers a natural set of test functions.

The Nyström method has a further generalization with a low-rank approximation of $k(Z, Z)$. Concretely, by letting $W_s$ be the best rank-$s$ approximation of $W = k(Z, Z)$ (given by eigendecomposition), we define the following $s$-dimensional kernel:

$$k_s^Z(x, y) := k(x, Z)W_s^+ k(Z, y). \tag{10}$$

Let $W = U\Lambda U^\top$ be the eigendecomposition of $W$, where $U = [u_1, \ldots, u_\ell] \in \mathbb{R}^{\ell \times \ell}$ is a real orthogonal matrix and $\Lambda = \mathrm{diag}(\lambda_1, \ldots, \lambda_\ell)$ with $\lambda_1 \geq \cdots \geq \lambda_\ell \geq 0$. Then, if $\lambda_s > 0$ we have

$$k_s^Z(x, y) = \sum_{i=1}^{s} \frac{1}{\lambda_i}(u_i^\top k(Z, x))(u_i^\top k(Z, y)). \tag{11}$$

So we can use functions $u_i^\top k(Z, \cdot)$ $(i = 1, \ldots, s)$ as test functions, which is chosen from a larger dimensional space $\mathrm{span}\{k(z_i, \cdot)\}_{i=1}^{\ell}$. Although closer to the original usage of the NyStöm method is to obtain $u_i^\top k(Z, \cdot)$ as an approximation of $i$-th eigenfunction of the integral operator $\mathcal{K}$ with $Z$ appropriately chosen with respect to $\mu$, we have adopted an explanation suitable for the machine learning literature [13, 34].

The following is a continuous analogue of Kumar et al. [34, Theorem 2] showing the effectiveness of the Nyström method. See also Jin et al. [26] for an analysis specific to the case $s = \ell$.

**Theorem 3.** *Let $s \leq \ell$ be positive integers and $\delta > 0$. Let $Z$ be an $\ell$-point independent sample from $\mu$. If we define the integral operator $\mathcal{K}_s^Z : L^2(\mu) \to L^2(\mu)$ by $f \mapsto \int_{\mathcal{X}} k_s^Z(\cdot, y)f(y)\,\mathrm{d}\mu(y)$, then we have, with probability at least $1 - \delta$, in terms of the operator norm,*

$$\|\mathcal{K}_s^Z - \mathcal{K}\| \leq \sigma_{s+1} + \frac{2\sup_{x \in \mathcal{X}} k(x, x)}{\sqrt{\ell}}\left(1 + \sqrt{2\log\frac{1}{\delta}}\right). \tag{12}$$

The proof is given in Appendix C.5. By using this estimate, we obtain the following guarantee for the random convex quadrature given by Algorithm 1 and the Nyström approximation.

**Corollary 4.** *Let $D_N$ be $N$-point independent sample from $\mu$ and let $Z$ be an $\ell$-point independent sample from $\mu$. Theorem 1 applied with the Nyström approximation $k_0 = k_{n-1}^Z$ yields an random $n$-point convex quadrature rule $Q_n$ such that, with probability at least $1 - \delta$ and $k_{\max} := \sup_{x \in \mathcal{X}} k(x, x)$,*

$$\mathbb{E}_{D_N}\left[\mathrm{wce}(Q_n; \mathcal{H}_k, \mu)^2 \mid Z\right] \leq 8\left(n\sigma_n + \sum_{m > n} \sigma_m\right) + \frac{16(n-1)k_{\max}}{\sqrt{\ell}}\left(1 + \sqrt{2\log\frac{1}{\delta}}\right) + \frac{2c_{k,\mu}}{N}.$$

*Proof.* From (11), $k^Z(x, y) - k_{n-1}^Z(x, y) = \sum_{i=n}^{\ell} \lambda_i^{-1}(u_i^\top k(Z, x))(u_i^\top k(Z, y))$ (ignore the terms with $\lambda_i = 0$ if necessary), and it is thus positive semi-definite. If we define $P_Z^\perp : \mathcal{H}_k \to \mathcal{H}_k$ as the projection operator onto the orthogonal complement of $\mathrm{span}\{k(\cdot, z_i)\}_{i=1}^{\ell}$, then, from (8), we also have $k(x, y) - k^Z(x, y) = \langle P_Z^\perp k(\cdot, x), P_Z^\perp k(\cdot, y)\rangle_{\mathcal{H}_k}$, so $k - k^Z$ is also positive semi-definite. In particular, $k - k_{n-1}^Z = (k - k^Z) + (k^Z - k_{n-1}^Z)$ is positive semi-definite. Also, it suffices to prove the result when $\sum_{m=1}^{\infty} \sigma_m < \infty$, so we can now apply Theorem 1.

For $k_1 := k - k_{n-1}^Z$, we prove the inequality $\int_{\mathcal{X}} k_1(x, x)\,\mathrm{d}\mu(x) = \sum_{m=1}^{\infty} \langle e_m, (\mathcal{K} - \mathcal{K}_s^Z)e_m\rangle_{L^2} \leq (n-1)\|\mathcal{K} - \mathcal{K}_s^Z\| + \sum_{m \geq n} \sigma_m$ (see (31) in Appendix D.2 for details), and the desired inequality follows by combining Theorem 1 and Theorem 3 (i.e., (5) and (12)). $\square$

**Remark 2.** *Algorithm 1 with the NyStöm approximation can be decomposed into two parts: (a) NyStöm approximation by truncated singular value decomposition (SVD) (the first $n - 1$ eigenvectors from an $\ell$-point sample), (b) Recombination from an $N$-point empirical measure. The complexity of (a) is $\mathcal{O}(n\ell^2)$, and it can also be approximated by randomized SVD in $\mathcal{O}(n^2\ell + \ell^2 \log n)$ [20]. The cost of part (b) is $\mathcal{O}(n\ell N + n^3 \log(N/n))$, where $n\ell N$ stems from the evaluation of $k_{1,\mathrm{diag}}$ for all $N$ sampling points. If we do not impose the inequality constraint regarding $k_{1,\mathrm{diag}}$, which still works well in practice, the cost of part (b) becomes $\mathcal{O}(\ell N + n^2\ell \log(N/n))$, by using the trick $\frac{1}{N}\sum_{i=1}^{N} U_{n-1}^\top k(Z, y_i) = U_{n-1}^\top \frac{1}{N}\sum_{i=1}^{N} k(Z, y_i)$, where $U_{n-1} = [u_1, \ldots, u_{n-1}] \in \mathbb{R}^{\ell \times (n-1)}$ is a truncation of the matrix that appears in the Nyström approxiamtion (10,11). So the overall complexity is $\mathcal{O}(n\ell N + n\ell^2 + n^3 \log(N/n))$ while an approximate algorithm (randomized SVD, without the inequality constraint) runs in $\mathcal{O}(\ell N + \ell^2 \log n + n^2\ell \log(N/n))$.*

## 2.4 Kernel Quadrature Using Expectations of Test Functions

Algorithm 1 and the bound (5) can be generally applicable once we obtain a low-rank approximation $k_0$ as we have seen in Section 2.2 and 2.3. However, since by construction we start by reducing the empirical measure given by $D_N$, it is inevitable to have the $\Omega(1/N)$ term in the error estimate and performance. We can avoid this limitation by exploiting additional knowledge of expectations.

Let $k_0$ and $k_1$ be positive definite kernels with $k = k_0 + k_1$. Let $\boldsymbol{\varphi} = (\varphi_1, \ldots, \varphi_{n-1})^\top$ be the vector of test functions that spans $\mathcal{H}_{k_0}$. When we know the expectations of them, i.e., $\int_{\mathcal{X}} \boldsymbol{\varphi}(x) \, \mathrm{d}\mu(x)$, we can actually construct a convex quadrature $Q_n = (w_i, x_i)_{i=1}^n$ satisfying

$$\sum_{i=1}^n w_i \boldsymbol{\varphi}(x_i) = \int_{\mathcal{X}} \boldsymbol{\varphi}(x) \, \mathrm{d}\mu(x), \qquad \sum_{i=1}^n w_i k_1(x_i, x_i) \leq \int_{\mathcal{X}} k_1(x, x) \, \mathrm{d}\mu(x) \qquad (13)$$

with a positive probability by an algorithm based on random convex hulls (Appendix D, Algorithm 2).

For this $Q_n$, we have the following theoretical guarantee (see Theorem 6 in Appendix B):

**Theorem 5.** *If a convex quadrature $Q_n$ satisfies the condition (13), then we have*

$$\mathrm{wce}(Q_n; \mathcal{H}_k, \mu)^2 \leq 4 \int_{\mathcal{X}} k_1(x, x) \, \mathrm{d}\mu(x).$$

If $k_0$ is given the Mercer/Nyström approximations, we immediately have the following guarantees; they correspond to **Mercer** and **Nyström** in Table 1. See also Theorem 9 and 11 for details.

- If $k_0(x, y) = \sum_{m=1}^{n-1} \sigma_m e_m(x) e_m(y)$ is given by the Mercer approximation, we have

$$\mathrm{wce}(Q_n; \mathcal{H}_k, \mu)^2 \leq 4 \sum_{m=n}^\infty \sigma_m$$

  for a convex quadrature $Q_n$ satisfying (13).
- Let $k_0 = k_{n-1}^Z$ be given by the Nyström approximation (10) with $Z$ being an $\ell$-point independent sample from $\mu$ (with $\ell > n$). Then, for a convex quadrature $Q_n$ satisfying (13), with probability at least $1 - \delta$ (with respect to $Z$) and $k_{\max} := \sup_{x \in \mathcal{X}} k(x, x)$, we have

$$\mathrm{wce}(Q_n; \mathcal{H}_k, \mu)^2 \leq 4 \left( n \sigma_n + \sum_{m > n} \sigma_m \right) + \frac{8(n-1)k_{\max}}{\sqrt{\ell}} \left( 1 + \sqrt{2 \log \frac{1}{\delta}} \right).$$

## 3 Numerical Experiments

In this section, we compare our methods with several existing methods. In all the experiments, we used the setting where we can compute $\int_{\mathcal{X}} k(x, y) \, \mathrm{d}\mu(y)$ for $x \in \mathcal{X}$ and $\iint_{\mathcal{X} \times \mathcal{X}} k(x, y) \, \mathrm{d}\mu(x) \, \mathrm{d}\mu(y)$ since then we can evaluate the worst-case error of quadrature formulas explicitly. Indeed, if a quadrature formula $Q_n$ is given by points $X = (x_i)_{i=1}^n$ and weights $\boldsymbol{w} = (w_i)_{i=1}^n$, then we have

$$\mathrm{wce}(Q_n; \mathcal{H}_k, \mu)^2 = \boldsymbol{w}^\top k(X, X) \boldsymbol{w} - 2\mathbb{E}_y[\boldsymbol{w}^\top k(X, y)] + \mathbb{E}_{y,y'}[k(y, y')] \qquad (14)$$

for independent $y, y' \sim \mu$ under $\int_{\mathcal{X}} \sqrt{k(x, x)} \, \mathrm{d}\mu(x) < \infty$, which is a well-known formula for the worst-case error [19, 58]. An essential remark shown in Huszár and Duvenaud [25] is that the Bayesian quadrature [49] with covariance kernel $k$ given observation at points $(x_i)_{i=1}^n$ (automatically) estimates the integral as $\sum_{i=1}^n w_i f(x_i)$ with $(w_i)_{i=1}^n$ minimizing the above expression. Once given points $(x_i)_{i=1}^n$ and additional knowledge of expectations, we can compute the optimal weights $(w_i)_{i=1}^n$ by solving a convex quadratic programming (CQP), either without any restrictions or with the condition that $(w_i)_{i=1}^n$ is convex. Although the former can be solved by matrix inversion, we have used the optimizer Gurobi[3] for both CQPs to avoid numerical instability. For the recombination part, we have modified the Python library by Cosentino et al. [11] implementing the algorithm of [61].

Our theoretical bounds are close to optimal in classic examples and we see that the algorithm even outperforms the theory in practice especially in Section 3.1. We also execute a measure reduction of a large discrete measure in terms of Gaussian RKHS and our methods shows a fast convergence rate in two ML datasets in Section 3.2. [4]

---

[3]Version 9.1.2, https://www.gurobi.com/

[4]All done on a MacBook Pro, CPU: 2.4 GHz Quad-Core Intel Core i5, RAM: 8 GB 2133 MHz LPDDR3.

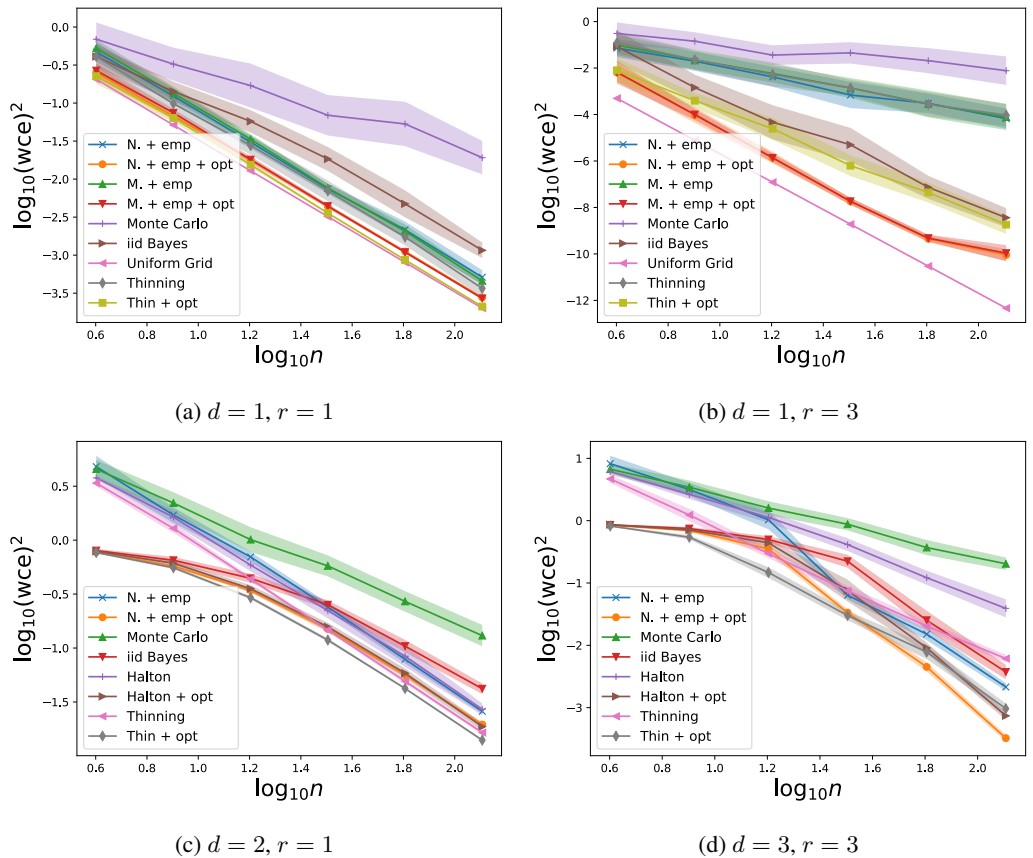

(a) $d = 1, r = 1$

(b) $d = 1, r = 3$

(c) $d = 2, r = 1$

(d) $d = 3, r = 3$

Figure 1: Periodic Sobolev spaces with kernel $k_r^{\otimes d}$: The average of $\log_{10}(\text{wce}(Q_n; \mathcal{H}_k, \mu)^2)$ over 20 trials is plotted for each method of obtaining $Q_n$. The shaded regions show their standard deviation. The worst computational time per one trial was 57 seconds of **Thin + opt** in $(d, r, n) = (3, 3, 128)$, where **Thinning** was 56 seconds and **N. + emp [+ opt]** was 22 seconds.

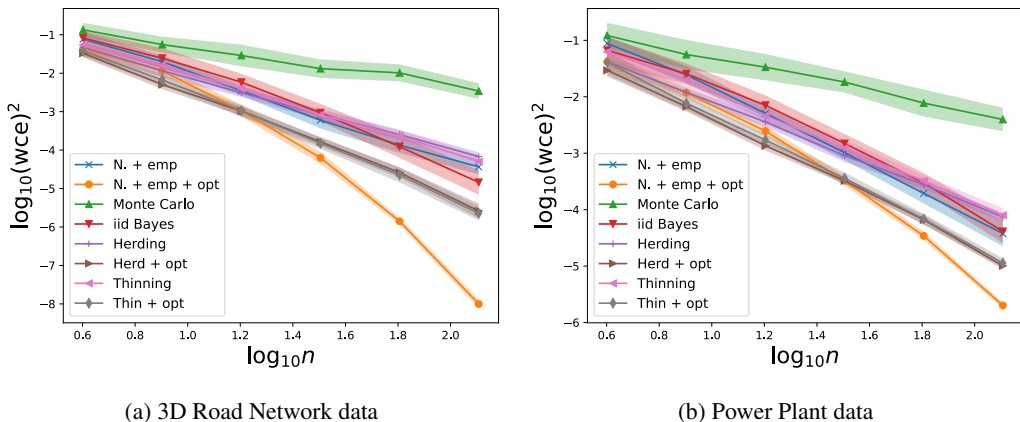

(a) 3D Road Network data

(b) Power Plant data

Figure 2: Measure reduction in Gaussian RKHS with two ML datasets: The average of $\log_{10}(\text{wce}(Q_n; \mathcal{H}_k, \mu)^2)$ over 20 trials is plotted for each method of obtaining $Q_n$. The shaded regions show their standard deviation. The worst computational time per one trial was 14 seconds of **Thinning [+ opt]** in Power Plant data with $n = 128$, where **N. + emp [+ opt]** was 6.3 seconds.

### 3.1 Periodic Sobolev Spaces with Uniform Measure

For a positive integer $r$, consider the Sobolev space of functions on $[0, 1]$ endowed with the norm $\|f\|^2 = (\int_0^1 f(x)\, dx)^2 + (2\pi)^{2r} \int_0^1 f^{(r)}(x)^2\, dx$, where $f$ and its derivatives $f^{(1)}, \dots f^{(r)}$ are periodic (i.e., $f(0) = f(1)$ and so forth). This function space can be identifies as the RKHS of the kernel

$$k_r(x, y) = 1 + \frac{(-1)^{r-1}(2\pi)^{2r}}{(2r)!} B_{2r}(|x - y|)$$

for $x, y \in [0, 1]$, where $B_{2r}$ is the $2r$-th Bernoulli polynomial [66, 3]. If we let $\mu$ be the uniform measure on $[0, 1]$, the normalized eigenfunctions (of the integral operator) are $1$, $c_m(\cdot) = \sqrt{2}\cos(2\pi m\cdot)$ and $s_m(\cdot) = \sqrt{2}\sin(2\pi m\cdot)$ for $m = 1, 2, \dots$, and the corresponding eigenvalues are $1$ and $m^{-2r}$ (both for $c_m$ and $s_m$). Although the rectangle formula $f \mapsto n^{-1}\sum_{i=1}^n f(i/n)$ (a.k.a. `Uniform Grid` below) is known to be optimal for this kernel [69, 47] in the sense of worst-case error, this RKHS is commonly used for testing the efficiency of general kernel quadrature methods [3, 7, 28]. We also consider its multivariate extension on $[0, 1]^d$, i.e., the RKHS given by the product kernel $k_r^{\otimes d}(\boldsymbol{x}, \boldsymbol{y}) := \prod_{i=1}^d k_r(x_i, y_i)$ for $\boldsymbol{x} = (x_1, \dots, x_d), \boldsymbol{y} = (y_1, \dots, y_d) \in [0, 1]^d$.

We carried out the experiment for $(d, r) = (1, 1), (1, 3), (2, 1), (3, 3)$. For each $(d, r)$, we compared the following algorithms for $n$-point quadrature rules with $n \in \{4, 8, 16, 32, 64, 128\}$.

**N. + emp, N. + emp + opt:** We used the functions $u_i^\top k(Z, \cdot)$ $(i = 1, \dots, n - 1)$ given by the Nyström approximation (11) with $s = n - 1$ as test functions $\varphi_1, \dots, \varphi_{n-1}$ in Algorithm 1. The set $Z$ was given as an $(\ell =)10n$-point independent sample from $\mu$. We used $N = n^2$ samples from $\mu$. In '**+ opt**' we additionally optimized the *convex* weights using (14)

**M. + emp, M. + emp + opt** $(d = 1)$**:** We used the first $n - 1$ functions of the sequence of eigenfunctions $1, c_1, s_1, c_2, s_2, \dots$ as test functions $\varphi_1, \dots, \varphi_{n-1}$ in Algorithm 1. We used $N = n^2$ samples from $\mu$. In '**+ opt**' we additionally optimized the *convex* weights using (14).

**Monte Carlo, iid Bayes:** With an $n$-point independent sample $(x_i)_{i=1}^n$ from $\mu$, we used uniform weights $1/n$ in **Monte Carlo** and the weights optimized using (14) in **iid Bayes**.

**Uniform Grid** $(d = 1)$**:** We used the rectangle formula $f \mapsto n^{-1}\sum_{i=1}^n f(i/n)$. This is known to be optimal [not just up to constant, but exactly; 69, 47], and thus equivalent to the Bayesian quadrature on the uniform grid, i.e., the weights are already optimized.

**Halton, Halton + opt** $(d \geq 2)$**:** For an $n$-point sequence given by the Halton sequence with Owen scrambling [21, 50], the uniform weights $w_i = 1/n$ is adopted in **Halton** and the weights are additionally optimized using (14) in **Halton + opt**.

**Thinning, Thin + opt:** Given an $N$-point independent sample $(y_i)_{i=1}^N$ with $N = n^2$ from $\mu$, an $n$-point subset $(x_i)_{i=1}^n$ taken from a KT++ algorithm (kernel thinning [14, 15] combined with Compress++ algorithm [56] with the oversampling parameter $\mathfrak{g} = \min\{4, \log_2 n\}$, implemented with GoodPoints package: https://github.com/microsoft/goodpoints) is adopted in **Thinning**. In '**+ opt**' we additionally optimized the *convex* weights using (14).

The results are given in Figure 1. In $d = 1$, the optimal rate given by **Uniform Grid** is known to be $\mathcal{O}(n^{-2r})$. As the uniform sampling is equal to the *optimized distribution* of Bach [3] in this case, **iid Bayes** also achieves this rate up to log factors. Although our theoretical guarantee for **M. + emp** is $\mathcal{O}(n^{1-2r} + N^{-1})$ with $N = n^2$ (Corollary 2), in the case $(d, r) = (1, 1)$, we can observe that in the experiment it is better than **iid Bayes** and close to the optimal error of **Uniform Grid**, but slightly worse than **Thinning**. Moreover, **N. + emp**, which does not use the information of spectral decomposition, is remarkably almost as accurate as **M. + emp** in $d = 1$. Furthermore, if we additionally use the knowledge of expectations, which **iid Bayes** is already doing, **M./N. + emp + opt** become surprisingly accurate even with $N = n^2$. They are worse than **Thinn + opt** when $r = 1$, but well outperform it when $r = 3$. Nonlineality in the graph of these methods when $(d, r, n) = (1, 3, 128)$ should be from numerical accuracy of the CQP solver (see also Section E.1).

The accuracy of **N. + emp + opt** becomes more remarkable in multivariate cases. It behaves almost the same as **Halton + opt** in $d = 2$ and clearly beats it in $d = 3$. Also, the sudden jump of our methods around $n = 30$ in $(d, r) = (3, 3)$ seems to be caused by the jump of eigenvalues. Indeed, for the integral operator given by $k_3^{\otimes 3}$ with uniform measure, the eigenspace of the largest eigenvalue

1 is of dimension 27, and the next largest eigenvalue is $1/64$. Again in the latter case, **N. + emp + opt** outperforms **Thin + opt**, and these results suggest that our method works better when there is a strong spectral decay, as is explicitly incorporated in our algorithm.

Note also that we can compare Figure 1 with Belhadji et al. [7, Figure 1] which includes some other methods such as DPPs, herding and sequential Bayesian quadrature, as we did experiments under almost the same setting. In particular, in the case $(d, r) = (1, 3)$ where the eigenvalue decay is fast, we see that our method substantially outperforms the sequential Bayesian quadrature.

### 3.2 Measure Reduction in Machine Learning Datasets

We used two datasets from UCI Machine Learning Repository (`https://archive.ics.uci.edu/ml/datasets/`). We set $\mu$ as the equally weighted measure over (a subset of) the data points $X = (X^{(1)}, \ldots, X^{(d)})^\top$ ($d = 3, 5$, respectively), where each entry is centered and normalized. We considered the Gaussian kernel $\exp(-\|x - y\|^2/(2\lambda^2))$ whose hyperparameter $\lambda$ is determined by *median heuristics* [17], and compared the performance of **N. + emp**, **N. + emp + opt** (with $\ell = 10n$, $N = n^2$), **Monte Carlo**, **iid Bayes**, **Thinning**, **Thin + opt**. We also added **Herding**, an equally weighted greedy algorithm with global optimization [10], and its weight optimization **Herd + opt** within *convex* quadrature given by (14). We conducted the experiment for $n \in \{4, 8, 16, 32, 64, 128\}$.

The first is **3D Road Network Data Set** [31]. The original dataset is 3-dimensional real vectors at $434874$ points. To be able to compute the worst-case error (14) efficiently to evaluate each kernel quadrature, we used a random subset $\mathcal{X}$ of size $43487 = \lfloor 434874/10 \rfloor$ (fixed throughout the experiment) and defined $\mu$ as the uniform measure on it. We determined $\lambda$ with the median heuristic by using a random subset of $\mathcal{X}$ with size $10000$ and used the same $\mathcal{X}$ and $\lambda$ throughout the experiment. The second is **Combined Cycle Power Plant Data Set** [32, 63]. The original dataset is 5-dimensional real vectors at $9568$ points. We set the whole data as $\mathcal{X}$ and defined $\mu$ as the uniform measure on it. We determined $\lambda$ with median heuristics by using the whole $\mathcal{X}$.

Figure 2 shows the results. We can observe that in both experiments **N. + emp + opt** successfully exploits the fast spectral decay of Gaussian kernel and significantly outperforms other methods. Also, even without using the knowledge of any expectations, **N. + emp** (and **Thinning**) show a decent convergence rate comparable to **Herding** or **iid Bayes**, which actually use the additional information. See also the end of Section E.2 for the plot of $\text{wce}(Q_n; \mathcal{H}_k, \mu')$ for another set of empirical data $\mu'$.

## 4 Concluding Remarks

We leveraged a classical measure reduction tool, recombination, with spectral properties of kernels to construct kernel quadrature rules with positive weights. The resulting algorithms show strong benchmark performance despite their restriction to convex weights. Our method has also recently been applied to Bayesian inference problems [1].

Although our method is applicable to fairly general situations, the usage or performance can be limited when it is difficult or inefficient to directly sample from the target measure $\mu$. Hence, an interesting follow up questions, is how one could replace the i.i.d. samples with smarter sampling (DPP, importance sampling, etc) before the recombination is carried out. Further, our theoretical results do not fully explain the empirical superiority; especially the $1/\sqrt{\ell}$ term does not match the experiments and it is a challenging future research question to reduce this theoretical gap. Nevertheless, we believe our method is the first generally applicable algorithm with a guarantee from the spectral decay.

## Acknowledgments and Disclosure of Funding

The authors would like to thank Chris Oates and Toni Karvonen for helpful remarks and discussions. The authors are also grateful to anonymous reviewers for detailed and constructive discussions that improved the paper. Harald Oberhauser and Terry Lyons are supported by the DataSıg Program [EP/S026347/1], the Alan Turing Institute [EP/N510129/1], the Oxford-Man Institute, and the Hong Kong Innovation and Technology Commission (InnoHK Project CIMDA).

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
