# OpenReview forum: "Positively Weighted Kernel Quadrature via Subsampling"
_NeurIPS.cc/2022/Conference — NeurIPS 2022 Accept_

### Official Review · Reviewer_bzNP · 2022-07-10

**Rating:** 6
**Confidence:** 3
**Soundness:** 1 poor
**Presentation:** 1 poor
**Contribution:** 2 fair

**Summary:**

The paper addresses a problem of kernel quadrature rules and proposes a new quadrature with positive weights that leverages recombination algorithms (stemming from Caratheodory’s Theorem) and either Mercer’s or Nystrom approximation of the kernel. Based on each version, there is also a derivation of corresponding convergence rates. The proposed methods are empirically verified on a classical setup for testing efficiency of general kernel quadrature methods.


**Questions:**

What are the consequences of treating a less general quadrature problem without weighting function $g$?

Could authors elaborate more on the need of convex quadrature? And how the positiveness of weights affects practical performance?

How the choice of recombination algorithm affects the performance?


**Limitations:**

The potential negative societal impact is not applicable to this submission.

The paper addresses a less general case of kernel quadrature, i.e. there is no weights function $g$ in the problem formulation (also stated in the related literature paragraph).

The method implies an ability to sample datapoints from target measure for recombination technique, however this could be problematic (this limitation is also stated by authors).


**Strengths And Weaknesses:**

Originality:

The paper addresses an established task of kernel quadratures. The method is based on well-known techniques of recombination and spectral kernel approximation combined in a unexplored way to obtain convex quadrature method. The paper emphasizes the major difference of their method being th convex weights of the quadrature points.

Quality:

I’m not entirely convinced by the very brief motivation for the convex weights, perhaps it would benefit the paper to elaborate the need for the convex quadrature in a more comprehensible way.

It seems that extensive supplementary material provides a substantial theoretical support for the proposed method, unfortunately, I was not able to check that in detail.

The Experimental Section lacks explicit comparison with the other practical methods apart from the worst and best case baselines on a known setting with Sobolev and Korobov spaces, e.g. there is no DPP-based method in the Figures.

It is also not immediately clear whether convexity of the weights has a crucial role in any performance advantages, thus the motivation for convex quadrature rules lack empirical support (at least in the experiments conducted and the conclusions drawn from their results).

The only method that is consistently bringing advantages across all Figures is +opt variation which assumes knowledge of expectations andadditional optimization over the weights, however Table 1 does not include this information which brings a little bit of a confusion. I wish that was a little bit more clearly stated in the paper.

Clarity:

There is a general issue with exposition, at least for someone less acquainted with the area. Abstract as well as Related Work would greatly benefit from a more extended exposition. As mentioned above, the motivation for the convex weights is also very short and could be made more clear with more explanations. Experimental Section doesn’t make it any clearer in this regard too, perhaps adding an ablation study coud offer more transparent understanding. The paper doesn’t discuss how their derived bounds compare against other methods, even under different assumptions, perhaps there should be matching setups allowing comparison.

Significance:

It is definitely an interesting method, and perhaps will raise a good amount of interest in the kernel community. However, I’m not sure whether the paper makes a good work at demonstrating the advantages of the proposed approach over existing methods. It first argues that convex quadrature is essential, but later fails to confirm this empirically, unfortunately, and it seems positive weights do not play a major role in theoretical analysis neither.

---

> ### Author Response · Authors · 2022-08-02
> **Response to Reviewer bzNP**
>
> > to elaborate the need for the convex quadrature in a more comprehensible way
>
> We give three different motivations in Section 1 for convex weights, Appendix B.3. We are also not the first to draw attention to the advantages of convex weights, see e.g. the cited reference Bach [2, Section 3.1]
>
> > supplementary material provides a substantial theoretical support for the proposed method, unfortunately, I was not able to check that in detail.
>
> That is of course fine, but one of your criticism (addressed below) is that convex weights do not play a major role in the analysis. If you look at the proofs in the appendix you will see that convexity is essential; many of the results are trivially wrong for non-convex weights.
>
> > Experimental Section lacks explicit comparison with the other practical methods
>
> We compare against the Monte-Carlo baseline, Quasi-Monte Carlo (Halton), and Bayes quadrature; further we added kernel thinning in the revised version. As we emphasize in the main text, Figure 1 is directly motivated by Belhadji et al. [6, Figure 1] which includes other methods such as DPPs and herding. In particular, since none of these come close to best performance we don't think there's much point in overcrowding the plot.
>
> > argues that convex quadrature is essential, but later fails to confirm this empirically, unfortunately, and it seems positive weights do not play a major role in theoretical analysis neither.
>
> To clarify, we never wrote that convex weights are essential but that for many natural situations, convex weights are advantageous. Regarding the experiments: these show something stronger, namely that even in situations where convex weights are apriori not required (e.g. no RKHS misspecification, no iteration of quadrature, etc) the proposed methods often outperforms others which are not limited to convex weights on standard benchmarks. Regarding the theoretical analysis: no, positive weights are essential for choosing points in our proofs that make the main part of the appendix.
>
> Once we have chosen points for use, considering only convex quadrature just limits the performance in terms of WCE, as the general quadrature (without any constraint on weights) simply includes convex quadrature and so never gets "beaten" by convex quadrature. However, limiting to convex quadrature during the algorithms, where we are still choosing a "sparse" set of points for use, we can greatly benefit from considering only convex quadrature thanks to its being probability measures as well as existence of efficient algorithms such as recombination.
>
> > consequences of treating a less general quadrature problem without weighting function?
>
> We would say that they are slightly different problems, as classical "quadrature" concerns the approximation of measures (i.e. a set of points with *specified weights*) and not for weighted integration for each weight function, where we have to determine the weights each time. Indeed, "kernel quadrature" has also been used for meaning this sort of integration rule without weights (e.g., Fuselier et al [2014; https://link.springer.com/article/10.1007/s00211-013-0581-1 ] or basically most of the papers other than Bach [2] or DPP-based kernel quadratures [5, 6, 7]). So the word "kernel quadrature" used by Bach [2] or Belhadji et al [6] is more like choosing "interpolation nodes" (indeed Belhadji et al [7] uses the word "kernel interpolation" for the same problem), although "kernel interpolation" also has a slightly different meaning [Wilson and Nickisch, ICML 2015, http://proceedings.mlr.press/v37/wilson15.html ].
>
> One major difference (relevant to our specific discussion here) is that for the latter case, where we only choose points (quadrature nodes), there seems to be no point in thinking about "convex quadrature" or "approximation by a probability measure" or whatever regarding the weights, as we have to change our weights according to the weight functions. So from the viewpoint of "kernel quadrature with weighting function", our method would look strange, though we want to emphasize that using discrete probability measures for approximating probability measures itself is quite common in the literature of numerical integration.
>
> Also, for the quadrature with weighing function, we additionally have to assume that we know the exact integration of e.g. (weighting function) * k(., x) or something, which is not a practical assumption. So the latter problem is good for theoretical assessment of whether the set of points well capture the distribution, but not generally leads to practical algorithms for the whole problem of "kernel quadrature with weighting function". Our (unweighted) problem setting is less general, but also thinking about the practicality of the algorithms. So we may say this "less general quadrature problem" is a practical restriction.

---

> ### Author Response · Authors · 2022-08-02
> **Response to Reviewer bzNP continued**
>
> Thanks for your time spent on our paper. We hope to convince you to reconsider your evaluation based on our answers and revision of the paper.
>
> > elaborate more on the need of convex quadrature
>
> We are happy to do so but it would be helpful if you describe which part of the three motivations we give in the introduction or the details in Appendix B.3 are unclear to you. Further, we emphasize that the advantages of convex weights are well-known in the kernel quadrature literature, see the reference Bach [2].
>
> > "choice of recombination algorithm affects the performance"?
>
> The choice of the algorithms itemized in our Remark 1 does not affect our theoretical guarantee regarding the convergence rate, but it affects computational complexity; some have better worst-case complexity bounds but in practice are outperformed by other methods. It is also possible that some sort of randomization can affect practical performance (like symmetrization in kernel thinning yields unbiasedness), and it is an interesting but challenging question requiring further study. Any progress on the classical recombination problem has the potential to further improve the complexity of our proposed method.

---

### Official Review · Reviewer_8cVE · 2022-07-12

**Rating:** 6
**Confidence:** 4
**Soundness:** 3 good
**Presentation:** 2 fair
**Contribution:** 2 fair

**Summary:**

The authors investigate the properties of kernel quadrature rules with convex weights. The authors proposed new algorithms that take as input i.i.d. samples, a suitable kernel, and provide a small integration error in the worst-case over the associated reproducing kernel Hilbert space. Two strategies are proposed based on kernel approximations via Mercer decompositions and Nystrom approximations along with recombination algorithms for point measures.



**Questions:**



I have a few major concerns:

1. For their Nystrom strategies (which are generically feasible for implementation), it is unclear if the mentioned bound can provide a better than Monte Carlo rate since that needs n / sqrt(l) < 1/ N which in turn means l > n^2 * N --- which is necessarily larger than N. On the other hand, the Mercer strategy in this work has a key limitation as it requires the knowledge of eigenfunctions, which is unknown for almost any generic (k, P) pair (other than a few exceptional pairs). Can the authors provide a detailed discussion about this aspect---as to the achieved rate in terms of the total number of queries---and provide concrete examples where their methods provide an improvement over prior strategies? In particular, further unpacking of Table 1 is needed, and it is unclear to me if the proposed strategy (E.g., see Table 2 https://arxiv.org/pdf/2105.05842.pdf].)

2.  This work is missing relevant references on recent new developments on Kernel thinning (KT)---which would have a tick on all C, M, E, in Table 1 of this work. Moreover, KT provides an error of order 1/n^2 + 1/N (in terms of the notation of this work) up to logarithmic factors for the WCE with n point output, and N point input (typically referred to as MMD in the literature). See [Table 2 https://arxiv.org/pdf/2105.05842.pdf], [Table 3 https://arxiv.org/pdf/2110.01593.pdf] for error rates, and [Example 6 https://arxiv.org/abs/2111.07941] for the new variant that provides near-linear runtime (N log^3 N), which is significantly faster than the strategies mentioned.

3. [Sec 1.2 https://arxiv.org/pdf/2105.05842.pdf] (KT paper) would also be a useful reference for some other works like P-greedy, black box importance sampling, etc. which are not referenced here and are relevant for the work.

4. For experiments, I believe KT should be added. Moreover, like the strategies in this work, even KT and herding output should be combined with 'opt' to provide a fairer comparison to their best-performing variant (N.+emp.opt].

5. At least one of the algorithms from Remark 1 should be discussed with some brief intuition for the readers to become aware of the high-level ideas used.

6. What is the probability in Thm 3 / Corr 4 taken over (is it the draw of Z)?

Minor comments:

- l 38/39: A reference is needed.


**Limitations:**

See questions.

**Strengths And Weaknesses:**


Strengths of the work include (a) perhaps a novel combination of two concepts: Spectral properties of kernels, and recombination results about point measured, and (b) the ability to simultaneously analyze strategies involving Mercer decomposition, and Nystrom approximations.

Limitations are (a) lack of clear discussion on when the proposed methods improve over prior strategies, while a summary is provided in Table 1, it is unclear if the methods indeed provide an improvement, (b) missing prior works, and (c) lack of stronger baselines that can be used in experiments. (see questions for more details),

---

> ### Author Response · Authors · 2022-08-02
> **Response to Reviewer 8cVE**
>
> Thanks for the specific questions and references. We have taken them into account as follows:
>
> > 1. Nystrom strategies (...) bound can provide a better than Monte Carlo rate
>
> For the appearance of $\sqrt{\ell}$ term, you are completely correct. This term doesn't appear in experiments and should be addressed by further theoretical developments. Nevertheless, as we have added in the conclusion section, we believe our method is meaningful as the first generally applicable algorithm with a bound based on spectral decay.
> For the eigenvalue decay in general, we have expanded on this in the contribution section. In short, whenever the eigenvalues $\sigma_n$ resp. the tail sum of eigenvalues $r_n$ decay quickly this gives a faster rate than MC. This is known for classic examples and our experiments (in particular Section 3.2) show that this seems to be fairly robust. Such decay conditions implicitly underpin many methods such as Nystrom that have found widespread use in applications, despite the fact that a characterisation of (kernel, measure) pairs that exhibit such a spectral decay seems currently out of reach.
>
> Related to this: yes, KT++ has better guarantees but in the experiments with strong spectral decay (Sobolev with higher order, or Gaussian kernels) it leads to worse performance than for example, our proposed method. We believe this is due to the fact that our approach directly bets on/exploits the spectral decay.
>
> > 2. missing relevant references on recent new developments on Kernel thinning (KT)
>
> Yes, thanks for these references! We have added KT both in Table 1 and have redone the experiments with KT, see Figure 1.
>
> > 3. useful reference for some other works like P-greedy, black box importance sampling, etc.
>
> Thanks! We have added these.
>
> > 4. For experiments, I believe KT should be added (...) herding output should be combined with 'opt' to provide a fairer comparison
>
> Thanks, we had missed both and now added them in the revision.
>
> > 5. algorithms from Remark 1 (...) high level ideas
>
> Thanks, we have added another paragraph in Remark 1.
>
> > 6. What is the probability in Thm 3 / Corr 4 taken over (is it the draw of Z)?
>
> Yes, it is taken over the draws of $Z$.

---

> > ### Comment · Reviewer_8cVE · 2022-08-06
> > **Second response**
> >
> > Thank you for your response, and for comparing with the additional baselines.
> >
> > I have some additional comments (I recently became aware of ref. in point 3 that seems very relevant for this work):
> >
> > 1. This continues to remain my main concern, and is related to my previous comment #1. The practical strategy proposed in this paper is Nystrom + Empirical for which the theoretical results offer no improvement over prior work. The strategies that offer theoretical improvement are not practical. I recommend the authors clarify these limitations in their contributions---e.g., after l27. This is an important point because there are numerous methods (including some baselines in this paper) that provide an improvement over Monte Carlo empirically while theoretically, the known bounds are no better than Monte Carlo. In fact, the term $n/\sqrt{\ell}$ for the practical strategy makes the theoretical result nowhere close to even being competitive with any other method--e.g., in several settings, the bound will be decaying as $n\sigma_n$ when the runtime is _exponential in $n$_. This point needs to be highlighted, and the primary contribution needs to be worded accordingly.
> >
> > 2. Can the authors provide examples when their bound of $n \sigma_n + r_{n+1}$ would decay for distributions beyond the unit cube (except Gaussian kernel Gaussian distribution case)? [That is would their results for the Nystrom strategy would provide any useful bound beside the simple examples?]
> >
> > 3. How does your work relate to this paper which analyzes the quality of Caratheodory coresets? https://arxiv.org/abs/2011.04907
> >
> >
> > 4. In Figure 2, to mimic the real-world settings, and the quantities that you are interested in, it would be better to plot the error results using a test set to construct a P. That is, the error is NOT measured with respect to the points that were compressed, but rather a fresh set of points. This would mean using another set of 43487 points in Fig 2(a) [since you have more data available], doing a data split for Fig 2(b) [since you used all the data], and using one-half of the data for compression, and the other half for measuring the integration error. Such a comparison would provide more trust in your experiments.

---

> > > ### Author Response · Authors · 2022-08-08
> > > **Second response to Reviewer 8cVE**
> > >
> > > Thank you for additional comments.
> > >
> > > > 1. I recommend the authors clarify these limitations in their contributions---e.g., after l27.
> > >
> > > Thanks, we explicitly added the paragraph  *Limitation* after the *Contribution* to clarify this point.
> > > We deferred the description of "intuition behind recombination" to Section B.6 accordingly due to the limitation of space.
> > >
> > > > 2. Can the authors provide examples when their bound of $n\sigma_n + r_n$ would decay for distributions beyond the unit cube (except Gaussian kernel Gaussian distribution case)?
> > >
> > > They are not readily available for general cases, but we can also say that our Nystrom approximation can make use of this implicit spectral decay. This said, we can say more about Gaussian kernels (than simply the case of Gaussian distribution).
> > > In Section B.3 (in the re-revised manuscript), we added a derivation of factorial decay of eigenvalues for Gaussian kernels with compactly supported distribution in $R$ (this can extend to multivariate cases, and sub-Gaussian distributions according to Bach [2, at the top of page 9] though we have not fully followed proofs).
> > >
> > > > 3. How does your work relate to this paper which analyzes the quality of Caratheodory coresets? https://arxiv.org/abs/2011.04907
> > >
> > > We were not aware of the paper but it is actually doing recombination for Fourier features (instead of Mercer or Nystrom features in our paper). The problem setting and proof techniques seem to be very different (function approximation in $L^2$ space, information-theoretic bound, and covering entropy based on given smoothness and compactness, etc..), but their naive use of Caratheodory's theorem 3.1.1 can be algorithmically greatly accelerated by using efficient recombination algorithms.
> > > We additionally mentioned it in the *Related Literature* section, thanks!
> > >
> > > > 4. In Figure 2, to mimic the real-world settings, and the quantities that you are interested in, it would be better to plot the error results using a test set to construct a P.
> > >
> > > Unlike the generalization analysis in usual machine learning tasks, we here simply have the triangle inequality regarding the MMD distance (also mentioned in Remark 1 of the KT paper, https://arxiv.org/abs/2105.05842), so we can predict the outcome quite well based on the plotted data and the size of empirical data we are using (*not* the magnitude of $N$ for recombination). But thank you for your suggestion, and we would add comments or experiments on this point in another revision to further improve the clarity of the paper.

---

> > > > ### Comment · Reviewer_8cVE · 2022-08-08
> > > > **Third response**
> > > >
> > > > I thank the authors for a speedy turnaround and a satisfactory response to my second set of comments.
> > > >
> > > > I will note one final comment about your response for 4: My comment was indeed about such a triangle inequality analysis---the gains (over the rate 1/n, e.g., over KT), that we see in Figure 2 would disappear once you use a test set since the error to the input points by your algorithm is likely way smaller than the error that will be introduced due to the test set of the same size. [I will assume it would be done in another revision as stated by the authors.]
> > > >
> > > > Once again, thank you for writing an interesting paper, and providing timely responses to my comments. I have increased my score accordingly.

---

### Official Review · Reviewer_vQx7 · 2022-07-13

**Rating:** 5
**Confidence:** 3
**Soundness:** 3 good
**Presentation:** 3 good
**Contribution:** 3 good

**Summary:**

This problem focuses on the quadrature problem, more specifically on bounding the maximum mean discrepancy between a quadrature approximation mu_Q= sum w_i delta_{x_i} and a target distribution mu, where the (xi) is a n-subset of an available bigger sample DN of i.i.d. samples of mu.
They focus on convex rules i.e. when the weights are positive and sum to 1.

The main idea is to consider a basis of n functions phi_1.. phi_n approximating RKHS functions, then sample N points (y1…yN), and select a n-subset (x1..xn) reweighted by (w1..wn). There are thus two sources of error (1) approximating k/the RKHS by a n-finite dimensional one, e.g. through Nystrom or Mercer approximations, (2) the empirical measure supported on y1…yN.

They provide several results, in particular th 1, which becomes more explicit in Cor 2 through Mercer approximation and Cor 4 through Nystrom approximation. The authors evaluate empirically their algorithm on small dimensional datasets (with either Nystrom or Mercer approximations) and compare it to the Monte Carlo rate , uniform grid and iid Bayes (weights minimizing equation 14) and demonstrate it achieves faster rates in practice than Monte Carlo.


**Questions:**

- Is it possible to clarify the dependence in number of samples in the quadrature n and dimension d in the bounds of Cor 1 and 2?
- can you report the slopes for various dimensions and infer a dependence dimension experimentally?


Minor comments
l55: “hybrid approaches” -> hybrid between what and what?
l69: a subset (xi)_{i=1}^n (n instead of N)
l71: (iii) in practice, constructing N>> n samples can be a challenge as well (eg MCMC methods are expensive)
l138: Nystom

**Ethics Review Area:**

["I don’t know"]

**Limitations:**

The limitations of the theoretical and empirical results could be discussed much more extensively, see comments above.

**Strengths And Weaknesses:**

Strengths
The paper is quite well written, and well referenced regarding kernel quadrature rules. It tackles an important problem, quadrature rules, which enables to approximate integrals accurately, with applications in Bayesian inference for instance.  The proposed algorithm has complexity O( nN + n 3 log(N/n) ) which is competitive with herding in high dimensions, where the latter become intractable because of the global optimization subroutine.

Weaknesses

Maybe I am missing something, but I think the bound in Theorem 1 does not advocate for faster rates than Monte Carlo.

In theorem 1, the expected squared MMD (where the expectation is taken wrt the available sample DN) is bounded by a constant c_1(n) (later on it will depend on n) + c_2/N.

Classical bounds on (squared) MMD (e.g. “A kernel two sample test”, Gretton et al.) between the empirical measure on DN (without quadrature weights) and the target mu is already of order c_2/N. Hence, from this bound it does not seem that the quadrature provides a better approximation of the target distribution mu, since if I understood well, Qn  (of size n \le N) can be of the same size than DN (size N) a priori.

The bound of Theorem 1 starts to be interesting if n is always much smaller than N and the term int(k-k0)dmu is much smaller than 1/n.
The first condition is typically satisfied (the authors claim l98 they will take N ~ n^2 ) later, so c_2/N becomes of order c_2/n^2, which is faster than Monte Carlo (c/n) . However, for the second term c_1(n)  it is less clear. Corollary 2 (Mercer approximation) yields a term c_1(n), indeed decreasing with n,  but it is not clear how fast c_1(n) decreases, faster than 1/n? Same question for Corollary 4 (Nystrom approximation) which is not very explicit in n.

if it is not faster than 1/n, then the quadrature rule (supported on n points) does not have a better upper bound currently than any empirical measure supported on n points of DN.

However, the experimental results show that the steep is lower (better) than the Monte Carlo rate, which encourages the method.
In the experiments, it would be interesting to report the steeps and understand the dependence of the rate with respect to n but also with respect to the dimension.

For these reasons, I tend to reject the paper, but I am ready to revise my opinion if the authors can clarify their results.

---

> ### Author Response · Authors · 2022-08-02
> **Response to Reviewer vQx7**
>
> Thanks for the feedback. We have addressed your comments as follows in the revision:
>
> > Maybe I am missing something, but I think the bound in Theorem 1 does not advocate for faster rates than Monte Carlo.
>
> Given a measure from which we can produce a large number $N$ of samples, our task is to construct a measure with support $n\ll N$ that integrates a class of functions approximately the same. In terms of the asymptotic $N \to \infty$ this can't do better than $1/N$. This has nothing to do with our specific method, but applies to any quadrature construction that requires $N$ samples, see eg. kernel thinning. The regime where such methods are interesting is $n\ll N$. You are completely correct, in general the bounds that we present do not beat MC but the spectral decay of the leading constants does counteract this. We have expanded on this in the contribution paragraph in Section 1. The intuition is that many measure/kernel pairs exhibit such a spectral decay and one can exploit it by the methods we propose; see also question 1 by reviewer 8cVE.
>
> >  clarify the dependence in number of samples in the quadrature n and dimension d in the bounds of Cor 1 and 2?
>
> > can you report the slopes for various dimensions and infer a dependence dimension experimentally?
>
> Thanks, it is a very reasonable question.
> It is rather about the interaction between dimensionality and eigenvalues $\sigma_n$ (or the "difficulty" of the problem), since our algorithm itself does not depend on the dimensionality of the problems.
>
> There should be several regimes for increasing the dimensiona of the problem; for example, as a multivariate version of (periodic) Sobolev spaces in our setting ($\sigma_n \sim n^{-2r}$ when $d=1$), we can not only consider a product RKHS aka Korobov spaces (in that case $\sigma_n \sim n^{-2r}(\log n)^{2r(d-1)}$), but also the classical multivariate Sobolev spaces (with $L^2$-norms with respect to all the partial derivatives up to degree $r$: then $\sigma_n \sim n^{-2r/d}$). These can be found in Bach [2, Section 2.3]. In the case of multivariate Gaussian kernel with Gaussian distribution, we have $\sigma_n\lesssim \exp(-cn^{1/d})$ for each dimension $d$. We will briefly add this remark when we get an additional page for the camera-ready version  (if applicable).
>
> > minor typos
>
> Thanks for these, all fixed.

---

### Official Review · Reviewer_pDdL · 2022-07-15

**Rating:** 8
**Confidence:** 5
**Soundness:** 3 good
**Presentation:** 4 excellent
**Contribution:** 3 good

**Summary:**

This article studies a family of quadrature rules suited for functions that belongs to an RKHS. The proposed construction is based on a recombination algorithm that takes a discrete measure $\nu_{N}$, that approximates the initial measure $\mu$ (for example, a Monte Carlo approximation),  and outputs another discrete  measure $\mu_{n}$. The weights of the quadrature are obtained by enforcing that the quadrature rule is exact on n functions $\phi_{1}, \dots, \phi_{n}$ that are taken to be equal to $\phi_{i} := k_{0}(x_{i},.)$ where $k_{0}$ is a ‘low rank’ kernel.

The contributions of the paper may be summarised as follows:
- A generic result (Theorem 1) that gives the upper bound of the worst-case error (on the unit ball of the RKHS) for the proposed algorithm for an arbitrary 'low rank' kernel $k_0$
- The instantiation of Theorem 1 to the case when $\nu_{N}$ is the Monte Carlo approximation of $\mu$ and $k_0$ is obtained from the Mercer decomposition of $k$
- The instantiation of Theorem 1 to the case when $\nu_{N}$ is the Monte Carlo approximation of $\mu$ and $k_0$ is obtained through the Nyström approximation
- Several numerical simulations that illustrate the theoretical rates.

The article is well written and the proven results were widely discussed and compared to the existing results in the literature.



**Questions:**


Questions and suggestions:

* In Section 3, it is not clear what you mean by the opt version: the optimization of (15) is done in the simplex or $R^{N}$?

* How do you explain the fact that in Corollary 2 the boundedness assumption (condition (a) in Theorem 8) is not required?

* It would be nice to add the N-th eigenvalue as a benchmark in the graphs

* Theorem 8 and Theorem 10 were mentioned implicitly in Table 1. In my opinion, I believe that they deserve to get an explicit mention in the main paper.

**Limitations:**

-

**Strengths And Weaknesses:**



Strengths:
* The empirical versions of the algorithms are flexible and may be used in domains where the eigenfunctions of the integration operator are not tractable.
* The proposed quadrature rules are convex, which is an important property in misspecified settings
* The theoretical analysis is insightful and may be used for other applications


Weaknesses:

* The empirical versions of the quadrature (when $\nu_N$ is Monte Carlo approximation of $\mu$) are very useful in practice, yet they come with weak convergence rates: the second term in the r.h.s. of the bound (5) in Theorem 1 is $\mathcal{O}(1/N)$, which is typically a slow rate in the kernel-based quadrature literature.

---

> ### Author Response · Authors · 2022-08-02
> **Response to Reviewer pDdL**
>
> Thank you for taking your time for reading our paper and reading even the appendix in detail. We would like to reply your comments below.
>
> > In Section 3, it is not clear what you mean by the opt version
>
> If we explicitly write "convex quadrature", the optimization of the weights is conducted over the simplex.
> So, in the revised manuscript, iid Bayes (= Monte Carlo + opt) and Halton are combined with the optimization over $R^n$,
> and other methods (N./M + emp, Thinning, Herding) are combined with the optimization over the simplex.
>
> > How do you explain the fact that in Corollary 2 the boundedness assumption (condition (a) in Theorem 8) is not required?
>
> Corollary 2 rather corresponds to (b) of Theorem 8. The randomness regarding satisfying the inequality without "boundedness assumption", that is qualitatively treated in Theorem 8(b), does not happen in the "+ empirical" case thanks to the deterministic nature of the recombination algorithm.
>
> > It would be nice to add the N-th eigenvalue as a benchmark in the graphs
>
> Yes, thanks for the suggestion! We tried it, but the graph is already pretty crowded and we ran out of time playing with the latex figure spacing to fit everything into 9 pages. We will try visibly improve the figures in the final version either by putting some plots in the appendix or otherwise.
>
> > Theorem 8 and Theorem 10 were mentioned implicitly in Table 1. In my opinion, I believe that they deserve to get an explicit mention in the main paper.
>
> Thanks, we agree but the main body is already quite dense. We should have enough space to explain these theorems and the `non-empirical' version of the algorithm if we have another content page in case of acceptance (if applicable).

---

> > ### Comment · Reviewer_pDdL · 2022-08-09
> > **Comment**
> >
> > Thank you for the response. I will keep my original score.

---

### Author Response · Authors · 2022-08-08
**Author update on the revision**

Thank you to all the reviewers for constructive comments and suggestions. Although we have already replied to each reviewer, we here summarize our primary updates of the revised manuscript in two parts:

- *Contribution and Limitation*: We have added two paragraphs after the paragraph starting from *Contribution.*
The first paragraph is about spectral decay, as it seemed to be better to provide more intuition on $\sigma_n$ or $r_n$, which are essential in kernel quadrature but not readily available from intuition. See also Section B.3 for a more general derivation of factorial decay in the case of Gaussian kernels. The second paragraph is clarifying the *Limitation* of our proposed methods, as we have a strong theory for Mercer-based methods which is not generally available (this limitation is also shared with other theory-oriented papers based on spectral properties), while our theoretical bound for Nystrom-based methods is not computationally competitive though are algorithmically practical and empirically perform very well.

- *Adding a strong baseline KT++*: Thanks to Reviewer 8cVE, we became aware of the competitive method called kernel thinning (https://arxiv.org/abs/2105.05842), and we added its recent variant (https://arxiv.org/abs/2111.07941) in our experiments. Interestingly, while KT++ or its '+ opt' outperforms most methods including ours in the case where spectral decay is moderate (= comparable to $1/n^2$), our methods becomes faster when there is a strong spectral decay (higher order Sobolev, or Gaussian RKHS). It empirically supports our explicit use of spectral decay via the Mercer/Nystrom approximation.

---

### Meta-Review · Area_Chair_A4TA · 2022-08-26

**Recommendation:** Accept
**Confidence:** Certain

**Metareview:**

We thank the authors and reviewers for their work throughout the reviewing process. The paper generated detailed and interesting discussions. While there remains minor concerns, we are confident that the paper brings new elements and will generate exciting discussions in the kernel quadrature community, and we are happy to recommend acceptance. We trust the authors to use all information in the discussion threads to polish the camera-ready version of the paper.

**Award:**

Yes

---

### Decision · Program_Chairs · 2022-09-14

Accept